# Exercise Modulates Brain Glucose Utilization Response to Acute Cocaine

**DOI:** 10.3390/jpm12121976

**Published:** 2022-11-30

**Authors:** Colin Hanna, John Hamilton, Kenneth Blum, Rajendra D. Badgaiyan, Panayotis K. Thanos

**Affiliations:** 1Behavioral Neuropharmacology and Neuroimaging Laboratory on Addictions, Clinical Research Institute on Addictions, Department of Pharmacology and Toxicology, Jacob School of Medicine and Biosciences, State University of New York at Buffalo, Buffalo, NY 14203, USA; 2Division of Addiction Research and Education, Center For Sports, Exercise and Global Mental Health, Western University Health Sciences, Pomona, CA 91766, USA; 3Department of Psychiatry, South Texas Veteran Health Care System, Audie L. Murphy Memorial VA Hospital, Long School of Medicine, University of Texas Medical Center, San Antonio, TX 78229, USA; 4Department of Psychology, State University of New York at Buffalo, Buffalo, NY 14203, USA

**Keywords:** rat, ^18^F-FDG fluorodeoxyglucose, positron emission tomography, aerobic exercise, glucose metabolism, Statistical Parametric Mapping, cocaine

## Abstract

Exercise, a proven method of boosting health and wellness, is thought to act as a protective factor against many neurological and psychological diseases. Recent studies on exercise and drug exposure have pinpointed some of the neurological mechanisms that may characterize this protective factor. Using positron emission tomography (PET) imaging techniques and the glucose analog [^18^F]-Fluorodeoxyglucose (^18^F-FDG), our team sought to identify how chronic aerobic exercise modulates brain glucose metabolism (BGluM) after drug-naïve rats were exposed to an acute dose of cocaine. Using sedentary rats as a control group, we observed significant differences in regional BGluM. Chronic treadmill exercise treatment coupled with acute cocaine exposure induced responses in BGluM activity in the following brain regions: postsubiculum (Post), parasubiculum (PaS), granular and dysgranular insular cortex (GI and DI, respectively), substantia nigra reticular (SNR) and compact part dorsal tier (SNCD), temporal association cortex (TeA), entopenduncular nucleus (EP), and crus 1 of the ansiform lobule (crus 1). Inhibition, characterized by decreased responses due to our exercise, was found in the ventral endopiriform nucleus (VEn). These areas are associated with memory and various motor functions. They also include and share connections with densely dopaminergic areas of the mesolimbic system. In conclusion, these findings suggest that treadmill exercise in rats mediates brain glucose response to an acute dose of cocaine differently as compared to sedentary rats. The modulated brain glucose utilization occurs in brain regions responsible for memory and association, spatial navigation, and motor control as well as corticomesolimbic regions related to reward, emotion, and movement.

## 1. Introduction

Cocaine use is a serious public health issue. Currently, there are more than 2 million people who regularly use cocaine in the United States [1]. Cocaine abuse is known to disproportionately affect marginalized populations and is associated with elevated hospitalizations and death [2]. Furthermore, cocaine addiction and abuse is known to lead to behavioral problems such as violence, complicating nearly all organ systems [3]. Health issues associated with cocaine use include an increased risk of stroke [4], dental issues and tooth decay [5], and cardiovascular complications [6]. Being a highly addictive substance, cocaine poses many challenges to clinicians seeking successful interventions that promote abstinence. Although psychological interventions for cocaine abuse such as cognitive behavioral therapy and contingency management have shown positive results, individuals seeking rehabilitation have a high dropout rate [1]. Supplemental interventions using both pharmacological [1] and non-pharmacological methods [7,8,9] are currently being observed in hopes of decreasing cocaine-addictive behaviors.

Among the non-pharmaceutical methods, aerobic exercise has been observed as both a preventative measure and a treatment option. In rats, chronic aerobic exercise has been found to decrease cocaine-seeking behaviors [10]. Aerobic exercise can also influence the brain’s reward systems [11]. Additionally, aerobic exercise has been found to activate brain areas involved in memory and sensory processing [12]. In humans, moderate levels of aerobic exercise are known to boost positive affect [13] which suggests potential in its rehabilitative qualities.

Positron emission tomography (PET) is an ideal neuroimaging technique for exercise neuroscience due to its ability to track longitudinally in the same subject regional and global metabolic changes, and its capacity to detail the integrity of neural activity [14]. FDG PET proved to be optimal in this experiment as it allowed us to compare configurations of brain activity between exercised and sedentary rats exposed to an acute dose of cocaine. Moreover, it allowed us to observe real-time brain metabolic changes due to acute cocaine exposure.

An acute dose of cocaine is known to evoke brain changes. In humans, it is known to decrease global cerebral blood flow [15]. In rhesus monkeys, a single dose of cocaine was shown to alter prefrontal BGluM and decline matching-task performance [16]. A single dose of cocaine has been shown to initiate immediate damage to prefrontal and hippocampal brain tissue in mice [17].

To our knowledge, there are no FDG PET studies that analyze the in-vivo effects of an acute cocaine dose following chronic exercise. There remains limited research on FDG PET and cocaine, and there is a lacking consensus about its metabolic consequences. In humans, cocaine abusers showed decreased frontal metabolism and a decrease in dopamine D2 receptor availability [18,19]. This pattern is consistent with the rhesus monkey data previously cited [16]. One study using male rats found that changes in BGluM levels are dependent on levels of addiction. More addicted rats showed a decrease in striatal and prefrontal metabolic activity, while less addicted rats showed an increase in BGluM in the caudate putamen and the medial pre-frontal cortex [20]. Genetically, the prevalence of the A1 allele is 50.9% in cocaine-dependent human subjects [21]. This was significantly higher than non-substance abusing controls (16.0%) (*p* < 10^−4^) and in population controls that did not exclude substance abuse (30.9%) (*p* < 10^−2^). A significant B1 allele prevalence (*p* < 10^−2^) was found in cocaine-abusing subjects compared to non-using controls. A regression analysis of regular cocaine users showed potent routes of cocaine use and an interaction of early deviant behaviors and parental alcoholism as risk factors that correlated with the A1 allele. These risk factors were positively (*p* < 10^−3^) correlated to A1 allelic prevalence, showing a strong association to the minor alleles (A1 and B1) of the dopamine receptor D2 gene (DRD2) with cocaine addiction suggesting that a gene located on the q22–q23 region of chromosome 11 confers susceptibility to cocaine addiction [21].

It is known that cocaine acts on the dopamine system, particularly in the basal ganglia. In mice, acute cocaine was found to decrease global BGluM. Particular decreases were found in the striatum, hippocampus, olfactory bulb, cerebellum, and thalamus compared to dopamine transporter knockout mice, in which only thalamic BGluM decreased [22]. PET studies have shown decreased glucose metabolism in brain regions of detoxified alcoholics and cocaine abusers. Reduced central dopaminergic function has been suggested in subjects who carry the A1 allele (A1+) compared with those who do not (A1). In a study by Noble et al., the mean relative glucose metabolic rate (GMR) was significantly lower in the A1+ than the A1- group in many brain regions, including the putamen, nucleus accumbens, frontal and temporal gyri and medial prefrontal, occipito-temporal, and orbital cortices [23]. Decreased relative GMR in the A1+ group was also found in Broca’s area, anterior insula, hippocampus, and substantia nigra [23].

The present study utilized a single dose of cocaine and FDG PET to look at changes in BGluM between chronically exercised and sedentary rats. These data could help us better understand the brain response to cocaine and how regular exercise may modulate this response.

## 2. Methods

### 2.1. Animals

Female Lewis rats at eight weeks of age (*n* = 13) (Taconic, Hudson, NY, USA) were housed individually. Room temperatures were held constant at 22.0 °C ± 2.0 °C. The lighting schedule was held on a 12-h reverse light/dark cycle (dark cycle: 8:00 a.m. to 8:00 p.m.). All animals were allowed food and water with no restrictions. All animals were handled daily to reduce the stress associated with handling. All experimental procedures were executed in compliance with the National Academy of Sciences Guide for the Care and Use of Laboratory Animals (1996). This experiment was approved by the University at Buffalo Institutional Animal Care and Use Committee**.**

### 2.2. Exercise Regimen

After one week of habituation, rats in the experimental group were subjected to forced exercise. A six-lane rodent treadmill was used for exercise. All rats in the exercise group (*n* = 6) underwent the same amount and intensity of running. The exercise paradigm was always performed during the rats’ dark cycle between the hours of 10:00 a.m. and 2:00 p.m. The exercise duration started at 10 min/day. The speed was held constant at 10 m/min. Only the running session times were increased by 10 min/day until a maximum of 60 min was run. A 10-min break from running was instated after the first 30 min. This forced exercise program continued for five days per week and concluded after six weeks, and the total distance run was approximately 16.5 km. Sedentary rats (*n* = 7) remained in their cages for the six weeks. An experimental timeline can be seen in Figure 1.

### 2.3. PET Imaging and Cocaine Injection

Fourteen days after exercise ended, all rats underwent acute cocaine dosing and PET scans. Food was restricted for 8 h prior to scans to normalize and control blood glucose levels. Rats were injected with 500 ± 115 μCi of ^18^F-FDG (intraperitoneal injection). The FDG injection was immediately followed by an acute dose of cocaine, which was dissolved in 0.9% saline. The solution was injected intraperitoneally at 25 mg/kg as previously described [8,10].

The uptake period lasted 30 min. After uptake, rats were anesthetized with isoflurane (3%, maintained at 1% for scan duration). Rats were secured to the bed of the scanner. PET scans lasted 30 min (as per standard imaging protocol). Scans were conducted using a PET R4 tomograph (Concorde CTI Siemens) as previously described [24]. Rats were returned to their home cages and given food and water after scans were completed.

### 2.4. Imaging and Statistical Analysis

Completed PET scans were first reconstructed via MAP algorithm (15 iterations, 0.01 smoothing value, 256 × 256 × 256 resolution). Manual co-registration with an MRI template (63 slices) [25] was carried out in PMOD imaging software (http://www.pmod.com, RRID:SCR_016547, version 2.85). Low quality PET images were omitted. PET images are included in Figure 2. MatLab Software was used for automatic co-registration and spatial normalization (http://www.mathworks.com/products/matlab/, RRID:SCR_001622 R2018b). Statistical Parametric Mapping software (SPM8) was used to identify regional changes in BGluM. Significant metabolic differences between the experimental and control group were found using a two-sample t-test (significant voxel threshold K > 50, *p* < 0.001). Significant BGluM clusters were viewed in PMOD imaging software (version 2.85, PMOD Technologies). Activation clusters are colored in hot scale, while inhibition clusters are colored in cold scale. Mapping and labeling were carried out utilizing the rat brain atlas [25].

## 3. Results

*t*-test results showed that six weeks of exercise and an acute dose of cocaine increased BGluM (*p* < 0.001, df = 11, K > 50) compared to sedentary rats in the following regions: postsubiculum (Post); parasubiculum (PaS); granular and dysgranular insular cortex (GI and DI); substantia nigra reticular (SNR) and compact part dorsal tier (SNCD); temporal association cortex (TeA); entopeduncular nucleus (EP); and crus 1 of the ansiform lobule (crus 1) (Figure 3, Table 1). Inhibition was observed (*p* < 0.001 df = 11, K > 50) in the ventral endopiriform nucleus (VEn) (Figure 3, Table 1). Complete details about cluster location, statistical significance, and voxel size can be seen in Table 1. Cluster locations and regions were used for brain mapping, presented in Figure 4.

## 4. Discussion

The reported data compares BGluM changes between chronically exercised and sedentary rats during a single cocaine dose, thus detailing the immediate effects of the drug and its associated short-term activation. The present preclinical study was identical to our previous work [12], save for the timing of the FDG scans and presence of an acute dose of cocaine. Interestingly, we can see some crossover between this research and what has been previously observed. First, we see a persistence of activation in the brain region(s) associated with auditory processing. Previously, we saw activation of the inferior colliculus and the primary auditory cortex. Presently, we found activation of the proximal TeA. Another repeated observation in this study included the activation of parahippocampal structures, the Post and the Pas. Lastly, the SNR and the SNDC suggest a regularity of basal activation after exercise, as our previous report found activation of the Caudate-Putamen [12]. These observations in FDG PET exercise neuroscience highlight the need for more standardized approaches in research.

The present study identified brain regions with BGluM activation (Post, PaS, GI, DI, SNR, SNCD, TeA, EP, and crus 1) and inhibition (VEn, 3), their baseline roles and functions, and the implications of this brain activity following chronic aerobic exercise and acute cocaine exposure. Ultimately, the aim of this research is to identify potential protective factors that aerobic exercise might offer against sudden cocaine exposure. We can speculate that the described changes might serve as a protective factor against the effects of cocaine due to past findings that suggest that aerobic exercise serves this function [7,8,10]. Past research confirms that aerobic exercise alters mesolimbic dopamine signaling and BGluM in regions involved in the “Brain Reward Cascade” [11,12,26]. This research seeks to map the metabolic changes that may be associated with the aforementioned neurochemical and behavioral changes.

Finally, we will discuss the hypothesized circuitry of these clusters in concert with each other. The proposed circuitry is based on the total results of the SPM analysis mapped on the rat brain and previous findings on neuropathways and anatomical connections. Functional connectivity has been previously reported with FDG PET imaging [12,24]

### 4.1. Parahippocampal Activation

Previously, we saw an activation of hippocampal subfields after chronic aerobic exercise [12]. We again saw activation of the Post coupled with activation of the PaS. These regions are encapsulated by the medial entorhinal cortex (MEC). Together, these three structures are essential for memory and spatial navigation [27,28]. This has been shown in three distinct type of cells that are densely packed in parahippocampal and MEC areas: grid cells, head direction cells, and border cells [29,30]. The function of head direction cells is self-evident: they activate with cranial orientation. Border cells fire when an animal is presented with spatial boundaries. Lastly, grid cells activate during environmental exploration and are believed to process location, distance, and direction [29,30,31]. Destruction of the Post is associated with impaired spatial memory and navigation [32]. Lesioning of the PaS shows similar results, including an impairment in object recognition [33]. Many studies show that exercise can have direct effects on the hippocampus and on memory [34,35].

The effects of cocaine on the hippocampus are unclear. Some studies indicate that cocaine (in certain contexts) can improve memory and function of the hippocampus [36,37]. Others suggest that cocaine, even at an acute dose, can cause neurodegeneration in the hippocampus and decrease neuronal proliferation [37,38]. The significance of the present hippocampal activity cannot be stated for certain, but previous reports have shown that exercise at varying intensities can influence susceptibility to cocaine reinstatement in rats [7,10], confirming the influence of both of these variables on the rodent memory systems.

### 4.2. Insular Cortex Activation

The insular cortex comprises a small portion of the cortical surface with a variety of functions such as perception and emotion [39]. It shares connections with sensory thalamic nuclei, the amygdala, the hypothalamus, and other limbic areas. [39,40]. The insular cortex is divided into three categories: granular (GI), dysgranular (DI), and agranular [25]. Our results find increased BGluM in the GI and DI after exposure to chronic aerobic exercise and acute cocaine.

Cocaine conditioning has been found to activate both the GI and DI in rats [41]. Additionally, inactivation of the GI has produced inhibition of drug-seeking behavior in rats [42,43].

The GI is involved in orofacial proprioception, including motor activation of the jaw [44]. Interestingly, cocaine has been reported to elicit oral alterations such as bruxism in humans [45], which poses a question about GI involvement in cocaine-related jaw tension and oral vasoconstriction. Perhaps the GI is activated in conjunction with the aforementioned parahippocampal areas that are so heavily involved in the animal’s sense of head direction, as many of our activated areas seem to play a role in cranial muscle movements. There are some data that suggest that the entorhinal cortex and the hippocampus share connections with the insula via the mediating amygdala, indicating that emotions are encoded with spatial information in this complex [46]. Additionally, retrograde and Cre-dependent anterograde tracing identified axonal projections from the parabrachial nucleus that extend to the insular cortex and the entorhinal cortex [47].

At first glance, the related literature might suggest that the timing and the type of aerobic exercise used in the present study may not offer protection against cocaine exposure via insular activation. However, in human studies the insular cortex responds with respect to exercise intensity. [48,49]. Follow-up studies might look to see how the insular cortex reacts to combinations of chronic cocaine and varying intensities of exercise.

### 4.3. Substantia Nigra Activation

The SNR and SNCD are a part of the substantia nigra (SN). The SN is a subdivision of the basal ganglia. The SN, particularly the compacta (SNC), features a high concentration of dopamine neurons that project into the striatum as seen in many Parkinson’s disease models [50,51]. The dopaminergic system in these regions plays a behavioral role in voluntary movement and addiction [52]. In rats, an acute cocaine dose has been found to alter dopamine receptor levels and glutamatergic inputs in the SN [53]. Additionally, the blocking of dopamine receptors in the SN significantly decreased the reward of cocaine self-administration [54]. It is, therefore, no surprise to see an increase in SN activity after an acute cocaine dose.

The SN and its connections, being highly dopaminergic, are heavily involved in decision making, reward processing, motivation, cognition, and responses to both rewarding and aversive experiences [55]. The dopaminergic systems in these regions are significantly involved in addiction and reward as they share dense dopaminergic connections with the ventral tegmental area and the nucleus accumbens, which together form a circuitry known as the Brain Reward Cascade [26]. The SN shares connections with prefrontal areas and motor areas, all of which are known to play a role in both decision and motor impulsivity [55].

The SN–striatal complex is essential for locomotion, as locomotor deficits are commonly associated with dopaminergic apoptosis of these neurons in Parkinson’s disease [52]. A single dose of cocaine resulted in the reverse effect by blocking the reuptake of dopamine, creating an immediate surplus in the synapse. However, a chronic decrease in dopamine levels is known to follow the intoxicating spike [56,57,58]. Withdrawal from cocaine is known to induce tremors and other movement disorders associated with dopamine depletion [59]. We might guess, therefore, that this SN activity could be dopaminergic, but we cannot be certain. Future studies could analyze dopamine levels and receptor availability after exposure to aerobic exercise and cocaine.

### 4.4. Activation of the Temporal Association Area

Previously, we observed an activation of auditory processing areas in exercised rats [12]. Though slightly altered, we again observed increased BGluM in a temporal region involved in the processing of auditory information: the TeA. The function of the TeA is in the localization of familiar stimulus [60] of all sensory inputs [61]. Circuitry between the previously activated primary auditory cortex and the presently observed TeA have been reported [60].

In rats, damage to the TeA is associated with deficits in visual-memory tasks [62]. In humans, many studies suggest its involvement in memory processing [61]. Damage to the TeA in humans is a common cause of various agnosias, including prosopagnosia—the inability to recognize faces [61]. It is known that exercise can improve memory functioning in humans [34] and act directly on neuroanatomy related to learning and memory in rats [35,62,63]. However, most of these studies on the effects of exercise and memory focus on the hippocampus. Additionally, it has been found that cocaine can enhance memory functioning in rats [36]. This evidence, coupled with increased activation in hippocampal regions, highly suggests that the present exercise/cocaine paradigm acts on the rat’s memory systems. We do, however, know that exercise alters the rodent’s associations, including associations to stimuli related to cocaine [7]. 

### 4.5. Entopeduncular Nucleus

The EP is considered to be the “rat equivalent” of the internal globus pallidus [64] and serves as an output of the basal ganglia [65,66]. One of the projections from the basal ganglia via the EP goes to the lateral habenula (LHb), an area known to actively respond to cocaine in rats [67]. Electrophysiology and retrograde tracer readings found that after an acute dose of cocaine, the EP activated the LHb more than any other afferent source. In the study by Li et al., the EP’s neuronal response to an acute cocaine dose occurred in two phases: an inhibitory phase associated with cocaine reward, and an excitatory phase that associated with the aversive effects of cocaine. When damaged, EP-LHb activation by cocaine is reduced. Additionally, avoidance behaviors commonly seen in the aversive phase of cocaine are attenuated after EP damage [67]. This provides insight into the crucial role of the EP in cocaine avoidance. However, it is unclear whether these PET results represent adverse or rewarding neurological effects. Future studies could investigate this by checking for stress in the rats with a corticosterone blood analysis as previously described [10].

To our knowledge, no known data exists to suggest that the EP responds to aerobic (or any other form of) exercise. However, the EP has been shown to contribute to locomotion and motor control [64,68]. Additionally, connections of the EP to the SN via the mediating nucleus tegmenti pedunculopontinus have been reported [69].

### 4.6. Crus 1 of the Ansiform Lobule

Crus 1, otherwise known as the superior semilunar lobule, a part of the cerebellum and is involved in cognitive and visuomotor functions, specifically eye movements, in many mammals [70,71]. Human studies suggest that the crus 1 responds to mechanical pain stimulation [72], and is also involved in cognition related to visual processing and eye movements such as reading [73,74,75] To our knowledge, there is no evidence to suggest that this area responds to cocaine alone. However, the crus 1 is known to receive ventral tegmental dopaminergic inputs [76], which is involved in motivation and addiction [77]. Therefore, we believe that increased metabolism of the crus 1 area is a result of cocaine exposure.

### 4.7. Inhibition of the Ventral Endopiriform Nucleus

Exercised rats showed decreased metabolic activity in the VEn. The function of this area is largely unknown. Some suggest that the VEn nucleus plays a crucial role in eleptogenesis [78]. Although we cannot distinguish the exact role of the VEn in response to cocaine or exercise, there are documented connections between the VEn and a few of our previously mentioned areas of activation, including the insular cortex and the subhippocampal/entorhinal areas [79,80]. Some reports have suggested that the VEn receives inputs from the insula and temporal cortex in cats [81], so it is possible that the activation of these areas is regulating the VEn inhibition.

More evidence is needed to fully define the role of the VEn in addiction and exercise neuroscience. However, the endopiriform nucleus has been shown to activate in response to oral pain and pressure induced in rats [80]. This, the activated GI, previously reported connections with the insula, and established dental problems associated with cocaine abuse in humans, prompts many questions into the neuromuscular and orthodontic physiopathology associated with an acute dose of cocaine. A full neurobehavioral FDG PET analysis would be warranted to further investigate this complex.

## 5. Conclusions

We see a clear difference in BGluM responses in chronically exercised rats compared to sedentary rats following an acute cocaine treatment. The BGluM responses were located in brain areas involved in motivation and addiction behaviors, as well as in muscle movement, spatial navigation, learning and memory. In addition, significant differences in BGluM were reported in regions responsible for cranial muscle control (activation of the GI/DI and inhibition of the VEn): both known to contribute to jaw muscle control. The Post/Pas are areas that play an important role in head direction. Lastly, the crus 1 in the cerebellum is known to have some input in ocular control and visuomotor tasks. These results give us insight to the BGLuM response following a single cocaine exposure and how regular exercise may impact this brain response. Modulation of BGluM in brain regions such as the TeA and the insula may explain some of the underlying protective factors that exercise has been shown to provide against drug-seeking behavior. Additionally, activation of mesolimbic areas and similarly connected cortical regions detail some of the brain areas associated with cocaine abuse and reward. These results might also provide some insight into the known orthodontic issues associated with cocaine use. Research continues on the benefits of exercise and drug abuse and neuroimaging data can help further identify and map neurocircuits and brain glucose utilization functional differences of the brain, such as the cortical and hippocampal activation post-exercise [12]. Previous research has confirmed the validity of these approaches and make deductions about the brain connectivity changes. Finally, we reported the BGluM changes and map of functional connectivity changes in aerobic exercise in response to acute cocaine exposure. This information will be very important for future assessments of how prescribed exercise may be incorporated as a tool to help recovery and relapse prevention in cocaine addicts [7,8,10,11,12,82,83,84,85,86,87,88,89,90]. This study looked at changes in brain activity in female rats. Future studies will include male animals. Additionally, future studies will look at various regimens of cocaine (including chronic cocaine exposure) and various intensities or prescribed doses of exercise. This could include high-intensity interval training or strength training.

## Figures and Tables

**Figure 1 jpm-12-01976-f001:**
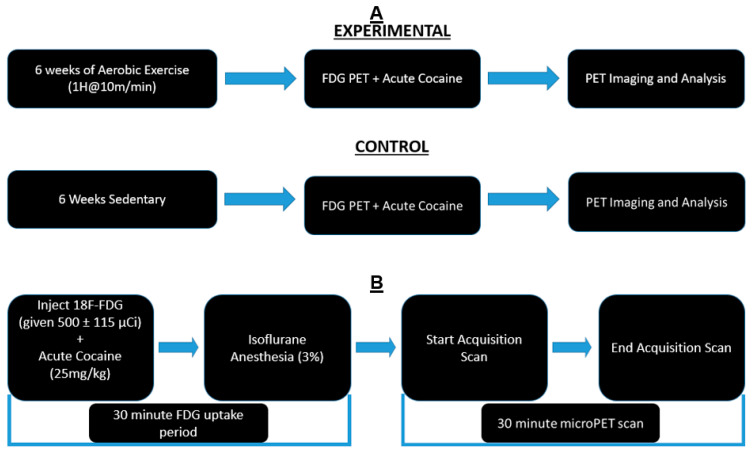
Experimental Timeline: (**A**) Animals were divided into exercise and sedentary groups. Exercise animals received 6 weeks of exercise, while sedentary animals remained in their home cages. All animals underwent PET scans and cocaine injections after the conclusion of 6 weeks. (**B**) Timeline of PET procedure: Animals were injected with [^18^F]-Fluorodeoxyglucose (FDG) and cocaine solution via intraperitoneal injection. They were returned to their home cages for a 30-min uptake period. At the end of the uptake period, animals were anesthetized and placed in the bed of the PET R4 tomograph machine. PET scans lasted 30 min. After the scan, animals were recovered and returned to their home cages.

**Figure 2 jpm-12-01976-f002:**
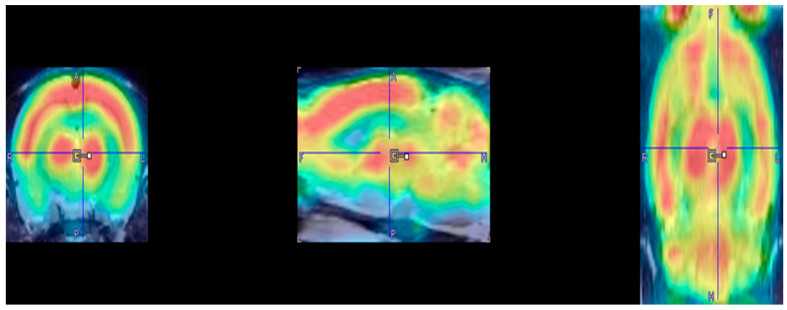
PET Images: Representative examples of a reconstructed PET image superimposed on an MRI template: coronal, sagittal, and horizontal (left to right).

**Figure 3 jpm-12-01976-f003:**
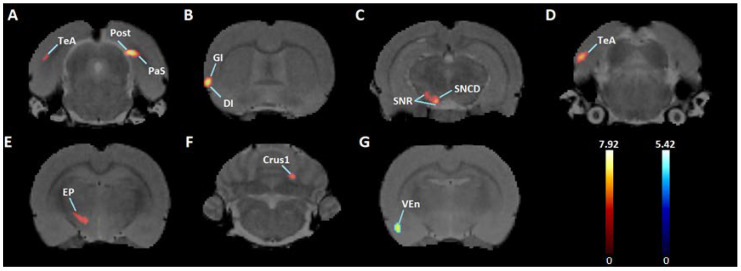
Significant Clusters: Coronal PET images showing brain regions with significant (*p* < 0.001, df = 11, and K > 50) metabolic activation (**A**–**F**) and inhibition (**G**) in exercised rats compared to sedentary rats. T-values represent peak activation (t = 7.92) and inhibition (t = 5.42). Hot scale clusters illustrate BGluM activation in the (**A**) Post and PaS; (**B**) GI and DI; (**C**) SNR and SNCD; (**D**) TeA; (**E**) EP; and (**F**) crus 1. Cold scale clusters represent inhibition, or a decrease in BGluM, in the (**G**) VEn.

**Figure 4 jpm-12-01976-f004:**
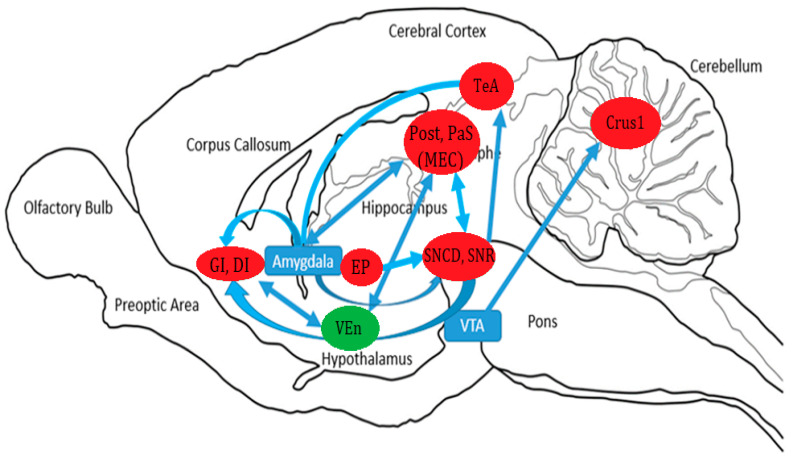
Hypothesized Circuitry: Sagittal drawing of hypothesized brain circuitry. Activated/increased BGluM clusters are shown in green. Inhibition of BGluM is shown in red. Blue boxes represent brain regions which act as relay points for our clusters.

**Table 1 jpm-12-01976-t001:** Brain regions with significant metabolic activity changes in exercised rats after an acute cocaine dose (*p* < 0.001 df = 11, K > 50). Cluster location is indicated by coordinates in stereotaxic space (medial–lateral, anterior–posterior, and dorsal–ventral). The t-values and z-scores were calculated from the average BGluM of all voxels within the significant clusters. KE represents the number of voxels in the respective cluster. Each cell under “Brain Region(s)” represents a separate cluster.

Brain Region(s)	Significant Effect	Medial-Lateral (mm)	Anterior-Posterior (mm)	Dorsal-Ventral (mm)	t-Value	z-Score	KE
Post PaS	+	3.6	−8.0	3.4	7.92	4.49	120
GI DI	+	−6.0	−0.4	6.6	7.92	4.49	311
SNR SNCD	+	−1.2	−5.6	8.2	5.85	3.86	64
TeA	+	−4.8	−7.6	3.4	4.14	3.15	132
EP	+	−1.8	−2.0	7.6	5.50	3.74	244
Crus 1	+	2.2	−11.6	4.2	4.98	3.53	64
VEn	-	5.2	−1.8	9.0	5.42	3.71	79

## Data Availability

Data is available from the corresponding author if needed.

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
