# Peer review of "Exercise Modulates Brain Glucose Utilization Response to Acute Cocaine"

_jpm, 2022, doi:10.3390/jpm12121976_

Round 1
Reviewer 1 Report
The study entitled "Exercise modulates brain glucose utilization response to acute cocaine" is novel and exciting research about the interaction between cocaine use and physical exercise. The authors focus their research on the ability of physical exercise to modify brain glucose metabolism after an acute dose of cocaine. In this sense, the data show important changes in different brain regions involved in a large number of tasks.
Regarding the methodology, a lack of information is observed in some components. It is recommended that the authors put the species of the rats, as well as explain the reason why only females have been chosen, and not have worked with males. Also 6 animals in the experimental group seem a bit tight. It is recommended to increase the N. The reason why a dose of 25mg/kg of cocaine is used is also not explained.
In the results section, the images have poor resolution. Likewise, the colors in figure 4 are contradictory when compared to figure 3, since in the latter figure the activations have been marked in red and the inhibitions in green, but in figure 4 the opposite has been done. It is recommended to mark in figure 4 the activated structures in red and the structure that undergoes inhibition in green.
The discussion is extensive but it is not fully explained why physical exercise generates this increase in activity in the regions described. It is also unclear whether this increased activity is a protective effect against the effects of cocaine. It would also be interesting to add future lines of research and the limitations of the study.
As a recommendation for researchers; the administration of different doses of cocaine, as well as different levels of physical exercise, would have made the study much more interesting and ambitious. Similarly, if these results would have differentiated between males and females, as appears to be the case in humans, would have provided much more information.
Author Response
Reviewer 1:
It is recommended that the authors put the species of the rats, as well as explain the reason why only females have been chosen, and not have worked with males. Also 6 animals in the experimental group seem a bit tight. It is recommended to increase the N. The reason why a dose of 25mg/kg of cocaine is used is also not explained. Species of the rat (Lewis Rat) has been added (2. Methods a. Animals, paragraph 1; pg 4). 25mg/kg equals one single dosing of cocaine in accordance to our past protocols and previous publications. Explanation and references have been added (2. Methods c. PET imaging and cocaine injection, paragraph 1; pg 5)
In the results section, the images have poor resolution. Image generation methods and resolution are the same as our past submissions (Hanna, Hamilton, Arnavut, Blum, & Thanos, 2022). These images can be re-submitted as separate files if needed. Original cluster image size: 1560x560px
Likewise, the colors in figure 4 are contradictory when compared to figure 3, since in the latter figure the activations have been marked in red and the inhibitions in green, but in figure 4 the opposite has been done. It is recommended to mark in figure 4 the activated structures in red and the structure that undergoes inhibition in green. This has been corrected, see figure 4 (page 7).
The discussion is extensive but it is not fully explained why physical exercise generates this increase in activity in the regions described. It is also unclear whether this increased activity is a protective effect against the effects of cocaine. Elaborated in discussion. See: 4. Discussion, paragraph 2, page 7: “We can speculate that the described changes might serve as a protective factor against the effects of cocaine due to past findings that suggest…”
It would also be interesting to add future lines of research and the limitations of the study. As a recommendation for researchers; the administration of different doses of cocaine, as well as different levels of physical exercise, would have made the study much more interesting and ambitious. Similarly, if these results would have differentiated between males and females, as appears to be the case in humans, would have provided much more information. Added, see last paragraph
Reviewer 2 Report
This paper reports the preliminary study to investigate the effect of aerobic exercise on acute exposure to cocaine using a preclinical study. The authors created an exercised group by running on the treadmills regularly and a control group without exercise. Both groups were rapidly exposed to a high dose of cocaine right after the FDG injection. Then, they checked the brain activities of both groups. Then, they compared the glucose consumption by the brains of both groups.
The reviewer feels that the authors used a very rough approach to demonstrate the effects of regular exercise. They could not explain their experimental results very well despite the extensive literature investigation. If they used more systematic approaches such as differences between low-dose and high-dose of cocaine and the degrees of exercise might explain the effects of the exercise more clearly. The reviewer also believes that the difference in the experimental setting might give some different results. Despite of the experimental approach, this result can be valuable because the study can create interest in the exercise effect on drug abuse.
The reviewer also points out to correct the following mistakes.
1. Please make 18 for superscript in lines Fig1, 19, 34, 141, and 147.
2. Line 62 --> The advantage of PET is not a good resolution, but a high sensitivity. If PET has a good resolution, PET imaging does not have to be merged with CT or MRI.
3. Line 93 --> Please write the full name of DRD2.
4. Line 115 --> Please indicate the species of the rats. Wistar? Sprague-Dawley?
5. Line 136 --> B should go to the next page.
6. Table 1 --> Lateral, Posterior, and Ventral should be in one line.
Author Response
This paper reports the preliminary study to investigate the effect of aerobic exercise on acute exposure to cocaine using a preclinical study. The authors created an exercised group by running on the treadmills regularly and a control group without exercise. Both groups were rapidly exposed to a high dose of cocaine right after the FDG injection. Then, they checked the brain activities of both groups. Then, they compared the glucose consumption by the brains of both groups.
The reviewer feels that the authors used a very rough approach to demonstrate the effects of regular exercise. They could not explain their experimental results very well despite the extensive literature investigation. If they used more systematic approaches such as differences between low-dose and high-dose of cocaine and the degrees of exercise might explain the effects of the exercise more clearly. We disagree, we explain our results and findings very clearly. The dose response to cocain was not the focus of this study. Instead, we chose a dose of cocaine that from the literature produces a behavioral response.
The reviewer also believes that the difference in the experimental setting might give some different results. This can be listed as a potential limitation, but again all rats were handled daily, weighed and housed in the same environment. The only difference is the treadmill running. In any case, non exercise clinical studies fo not require sedentary control subjects to stand in a stationary treadmill.
Despite of the experimental approach, this result can be valuable because the study can create interest in the exercise effect on drug abuse.
The reviewer also points out to correct the following mistakes.
- Please make 18 for superscript in lines Fig1, 19, 34, 141, and 147. Fixed 11/21
- Line 62 --> The advantage of PET is not a good resolution, but a high sensitivity. If PET has a good resolution, PET imaging does not have to be merged with CT or MRI.Removed from line 62- 11/21
- 3. Line 93 --> Please write the full name of DRD2.Corrected 11/21
- 4. Line 115 --> Please indicate the species of the rats. Wistar? Sprague-Dawley?Lewis: See 2. Methods, a. Animals, line 116
- 5. Line 136 --> B should go to the next page.B moved to page 5
- Table 1 --> Lateral, Posterior, and Ventral should be in one line.Not sure what is meant by this suggestion. Any ideas?
Hanna, C., Hamilton, J., Arnavut, E., Blum, K., & Thanos, P. K. (2022). Brain Mapping the Effects of Chronic Aerobic Exercise in the Rat Brain Using FDG PET. J Pers Med, 12(6). doi:10.3390/jpm12060860
Round 2
Reviewer 1 Report
Thanks for the corrections